# Cognitive Functions Associated with Brain Imaging Markers in Patients with Psoriasis

**DOI:** 10.3390/ijerph19095687

**Published:** 2022-05-07

**Authors:** Luiza Marek-Jozefowicz, Adam Lemanowicz, Małgorzata Grochocka, Monika Wróblewska, Katarzyna Białczyk, Katarzyna Piec, Grzegorz M. Kozera, Zbigniew Serafin, Rafał Czajkowski, Alina Borkowska

**Affiliations:** 1Department of Dermatology and Venerology, Faculty of Medicine, Nicolaus Copernicus University in Toruń, Ludwik Rydygier, Collegium Medicum in Bydgoszcz, 85-094 Bydgoszcz, Poland; malgorzatagrochocka@gmail.com (M.G.); r.czajkowski@cm.umk.pl (R.C.); 2Department of Radiology and Diagnostic Imaging, Faculty of Medicine, Nicolaus Copernicus University in Toruń, Collegium Medicum in Bydgoszcz, 85-094 Bydgoszcz, Poland; adam.lemanowicz@cm.umk.pl (A.L.); serafin@cm.umk.pl (Z.S.); 3Department of Health Economics, Nicolaus Copernicus University in Toruń, Collegium Medicum in Bydgoszcz, 85-094 Bydgoszcz, Poland; monika.wroblewska@cm.umk.pl (M.W.); katarzyna.bialczyk@cm.umk.pl (K.B.); 4Department of Neurology, Faculty of Medicine, Nicolaus Copernicus University in Toruń, Collegium Medicum in Bydgoszcz, 85-094 Bydgoszcz, Poland; katarzyna.piec@cm.umk.pl; 5Medical Simulation Centre, Faculty of Medicine, Medical University of Gdańsk, Dębowa 17 Street, 80-208 Gdańsk, Poland; gkozera@gumed.edu.pl; 6Department of Clinical Neuropsychology, Nicolaus Copernicus University in Toruń, Collegium Medicum in Bydgoszcz, 85-067 Bydgoszcz, Poland; alab@cm.umk.pl

**Keywords:** psoriasis, neuropsychological test, brain MRI correlates

## Abstract

Psoriasis is a severe inflammatory disease associated with a higher comorbidity of depression, cognitive dysfunction and brain atrophy. The association between psoriasis, magnetic resonance imaging (MRI) markers and cognitive impairment has rarely been investigated, and the existing results are conflicting. Methods. This study included 89 subjects (53 patients with psoriasis and 36 healthy controls). The severity of psoriasis was evaluated using the Psoriasis Area and Severity Index (PASI) score; for depression, the Hospital Anxiety and Depression Scale (HADS) scale was used. Neuropsychological tests were also applied, including a Trail Making Test (TMT) as well as Digit Span, Stroop, Verbal Fluency and Rey Auditory Verbal Learning tests. MRI scans were performed using a 1.5 T scanner. Brain volumetry, white matter lesions, grey matter and white matter were evaluated. The extent of these changes was assessed on the Fazekas scale. The differences between groups were evaluated using a Student’s *t*-test and a Mann-Whitney U test, and a Pearson correlation analysis was also performed. Results. Patients with psoriasis presented worse achievements on all the neuropsychological tests and showed more intense changes on MRI compared to healthy controls. The severity of psoriasis as determined by PASI scores was associated with depression, and a greater psychomotor slowness severity of changes in the brain was associated with poorer results on the neurological tests. Conclusions. Our results indicate the possibility of progressive brain atrophy related to cognitive decline in psoriasis.

## 1. Introduction

The brain and skin are anatomically and functionally connected, and they arise from a common embryonic origin—the ectoderm. Psoriasis is a chronic immune-mediated inflammatory skin disease [1,2] that affects 0.5–11.4% of adults and 1.4% of children worldwide [3]. Depression is a highly prevalent disease characterized by affective and cognitive disturbances [4]. Increasing evidence has shown that psoriasis can lead to depression, and depression, in turn, can exacerbate psoriasis, which may result in a vicious cycle of psoriasis and depression [5]. A study investigating 2391 psoriasis patients indicated that 62% of them had depressive symptoms [6]. Other studies have shown that depression generally predates psoriasis onset, and patients with moderate to severe depression have a significantly increased risk of psoriasis [7,8]. Therefore, there is a potential association between psoriasis and depression.

Psoriasis is characterized by an increased risk of developing atherosclerosis [9] and cardiovascular events [10,11,12], and the potentially early appearance of cognitive dysfunctions [13,14]. Some studies have suggested that psoriasis is associated with cognitive impairment and dementia [15,16,17]. The process is stepwise, starting with increased chronic neuroinflammatory cytokine levels that contribute to atherogenesis and microvascular injury [18].

Vasculopathy, including arterial stiffness and impaired endothelial function, may predispose patients with psoriasis to dementia, especially vascular dementia. Oxidative stress and pro-inflammatory cytokines, which are increased in patients with psoriasis, can interfere with synaptic plasticity and neurogenesis, promoting neurodegenerative processes and contributing to a decline in cognitive function [19,20,21]. In one study, cortical thickness analysis revealed a reduction in brain thickness in the parahippocampal gyrus, superior temporal gyrus and superior frontal gyrus of the left and right hemisphere. These neuroradiological findings have a direct correlation with the results of cognitive tests; the brain areas where thickness is reduced are those involved in cognitive functioning [15]. The mechanism linking psoriasis to cognitive impairment may be complex. It is possible that a common genetic background links psoriasis to Alzheimer’s disease.

The purpose of this study was to the evaluate MRI correlates of cognitive functioning in relation to age, clinical parameters and depression in patients with psoriasis.

## 2. Materials and Methods

### 2.1. Study Design

We conducted a cross-sectional MRI and neuropsychological study on hospitalized patients with a moderate severity of psoriasis symptoms compared to heathy controls. Initially, 55 psoriasis patients were recruited to the study; however, two of them dropped out—patient 1 because of their inability to perform the Color-Word Interference Test (colorblindness), and patient 2 due to their discharge from the hospital before the start of the examination. The dermatologist who performed the initial examination of the patients was blinded to the results of the cognitive tests performed by a neuropsychologist and neuroimaging data analyzed by a radiologist.

### 2.2. Patients

A total of 53 patients with chronic psoriasis, 20 female (37.7%) and 33 male (62.3%), aged 21–68 years (mean age 49.1 ± 12.8 years), volunteered for this study. The duration of illness was 14.51 ± 9.43 years, and the duration of the current episode of psoriasis was 4–9 weeks (mean of 7.64 ± 4.35 weeks). The intensity of psoriasis measured by the PASI scale was 18.84 ± 5.78 p. The intensity of depressive symptoms measured by the HADS scale was 15.47 ± 7.36 p.

The control group consisted of 36 healthy subjects, 15 female (41.7%) and 21 male (58.3%), aged 29–60 (mean age of 43.7 ± 11.06 years), without any history of psoriasis or other autoimmunological diseases, and also without substance abuse, neurologic disorders, anxiety and depressive symptoms or other psychiatric illnesses. The investigated group was matched by age, and no significant differences between comparison groups were observed (*p* = 0.393, Student’s *t*-test).

All investigated subjects provided written informed consent for participation in this project according to the approval of the Bioethics Committee of Collegium Medicum in Bydgoszcz, Nicolaus Copernicus University in Torun, Poland (Ref. No. KB 724/2017). Following the agreement of the Bioethics Committee at the Nicolaus Copernicus University, researchers and bank management, the study was conducted with complete anonymity of the participants and in line with the committee’s consent. This paper does not contain identifiable features of the patients. The study was conducted in accordance with the Declaration of Helsinki.

### 2.3. Methods

#### 2.3.1. Psoriasis and Depression

The severity of psoriasis was evaluated by a dermatologist. Patients receiving any systemic treatment for psoriasis, including acitretin, cyclosporine, methotrexate and phototherapy, for at least 3 months before enrolment were not included in the study. Patients affected by psoriatic arthritis, diagnosed according to the classification criteria for psoriatic arthritis, were not included in the study either. Any use of a drug able to cross the blood–brain barrier or any documented psychological or psychiatric disease resulted in exclusion. The examination and assessment of the severity of dermatological lesions using the PASI scale was conducted by one physician. A PASI score of more than 10 was considered moderate, while a score more than 50 was considered as severe disease [22,23]. The intensity of depressive symptoms was evaluated by a clinical psychologist using the Hospital Anxiety and Depression Scale [24,25].

#### 2.3.2. Neuroimaging

MRI scans were performed in the supine position. A 1.5 T scanner (Optima 450w, GE Healthcare, Waukesha, WI, USA) with a 32-channel head neck unit coil was used. High resolution T1-weighted and FLAIR scans (fluid-attenuated inversion recovery) were acquired. The T1-weighted scans were acquired with a 3-dimensional sagittal ultrafast gradient echo sequence under the following parameters: repetition time = 8.7 ms; echo time = 3.4 ms; inversion time = 450 ms; flip angle = 12°; acquisition matrix = 240 × 240; field of view = 240 mm; slice thickness = 1 mm; and total number of slices = 160. The scan duration was 3 min and 14 s. Then, the FLAIR sequence scans were acquired with a 3-dimensional sagittal fast spin echo sequence under the following parameters: repetition time = 8500 ms; echo time = 130 ms; inversion time = 2186 ms; acquisition matrix = 224 × 224; field of view = 256 mm; slice thickness = 1.2 mm; and total number of slices = 128. The scan duration was 6 min and 26 s.

Brain volumetry and white matter lesions (WML) were evaluated using VolBrain (https://volbrain.upv.es; accessed on 1 January 2018), a cloud-based automatic segmentation tool. The T1-weighted images were segmented into gray matter (GM) and white matter (WM) by a segmentation algorithm, which allowed us to determine the volume of individual brain structures (e.g., left and right hemispheres, cerebellum, etc.). Similarly, the FLAIR scans were uploaded to VolBrain for automated segmentation and volumetry of white matter lesions. Finally, two independent observers (trained radiologists with 10 and 5 years of experience) reviewed the FLAIR scans to assess the extent of the changes in deep and periventricular white matter on the Fazekas scale.

#### 2.3.3. Neuropsychological Testing

Trail Making Test (TMT) part A was used to evaluate psychomotor speed and visuospatial orientation, and TMT part B was employed for visuospatial working memory and set shifting evaluation. A Stroop Color-Word Interference Test (SCWIT) part A (reading words) was used for the assessment of attention and reading speed, and part B (naming of color of words when the color of a printed word is different than color defined by the word) was applied for the evaluation of executive functions and complex attentional processes. Memory processes were analyzed using a Digit Span (DS) test consisting of two parts: DS forward, which measures verbal memory capacity, and DS backward, which measures the ability to keep and manipulate information in short term memory—a process that involves working memory.

To evaluate learning and memory processes, a Rey Auditory Verbal Learning Test (RAVLT) was used, with five sets of immediate recall and one set of delayed recall. Verbal fluency was assessed using a Verbal Phonological Fluency Letter Test (VF).

#### 2.3.4. Statistical Analysis

All data were analyzed statistically with IBM SPSS Statistics. The normality of variable distributions was checked using a Shapiro-Wilk test. Depending on the variable distribution, the mean value and standard deviations (SDs), median values and IQR 25–75%, or frequencies were calculated. The differences between the groups of patients were analyzed using a Student’s *t*-test for independent variables and a Mann-Whitney U test. Correlations between the analyzed parameters were verified using a Pearson correlation test (*p*-correlation coefficient). The results were considered statistically significant if the *p*-value was less than 0.05.

## 3. Results

Table 1 presents the performance of the investigated subjects on the neuropsychological tests. As shown in Table 1, the performance on all neuropsychological tests applied was significantly worse in psoriasis patients compared to healthy controls.

Table 2 presents the results of the neuroimaging parameters in psoriasis patients. 

### Correlation Analysis

The severity of psoriasis was positively correlated with the intensity of depressive symptoms (R = 0.46, *p* = 0.01) and the time of performance on the TMT A (R = 0.43, *p* = 0.02). Depressive symptoms were also correlated with performance on the TMT A (R = 0.43, *p* = 0.02). This indicated that patients with more severe psoriasis, especially in old age, may develop more severe depression and psychomotor slowness. No association between the other clinical parameters and variables analyzed were found. Older age was found to be associated with brain impairment and cognitive decline in psoriasis patients (Table 3).

The correlations between cognitive performance and neuroimaging variables in subjects with psoriasis are shown in Table 4. The time taken to complete the TMT A and B tests was positively correlated with WML volume; TMT A also showed a positive correlation with periventricular WML on the Fazekas scale. Better results on the Digit Span backward test, which assesses working memory, were correlated with higher grey matter volume. Stroop A test scores, measuring mostly attention, were associated with deep WML volume, Fazekas deep WML volume and Fazekas periventricular WML scoring. Verbal Fluency test performance was positively correlated with brain volume and grey matter volume and negatively correlated with deep WML volume. More robust correlations were found between the neuroimaging variables and performance on the RAVLT. Higher scores (i.e., better performance) of most of the AVLT parameters were negatively correlated with total WML volume, deep WML volume, Fazekas deep WML count, periventricular WML volume and Fazekas periventricular WML count. Positive correlations were observed between better performance on the AVLT and higher scores for brain volume, cerebellum volume and grey matter volume. 

Because the most correlations were found between the MRI parameters and the efficiency of verbal memory delayed recall on the RAVLT, a multivariate regression model was built. The proposed two-step model used progressive selection (criterion: probability of F-introduction ≤ 0.050), which allowed for the inclusion of two variables: brain volume (B= −3.11; standard error = 1.005; Beta = −0.45; t = −3.095; significance = 0.004) and Fazekas periventricular WML (B = 0.013; standard error = 0.006; Beta = 0.295; significance = 0.050). This model exhibited a variance of about 32% (R^2^ = 0.32). The regression plot of standardized residues for variable AVLT delayed recall is presented in Figure 1.

## 4. Discussion

The results from the neuropsychological tests indicated poorer cognitive abilities in patients suffering from psoriasis compared to healthy control subjects matched for age, education and gender. The performance on all neuropsychological tests applied was significantly worse in psoriasis patients compared to healthy controls. Patients required a longer time to complete Trail Making Test A and B, which indicates impairments in psychomotor speed, spatial orientation and set shifting. They also showed worse achievements on all items of the Rey Auditory Verbal Learning Test, which indicates poorer memory and learning processes, including delayed memory. The worse results on the Stroop Color-Word Interference Test part B reflects impairments in verbal working memory and executive functions. Patients with psoriasis also presented with a slight weakness in verbal fluency compared to healthy subjects. On the Digit Span test, no significant difference was observed between groups. The overall cognitive profile suggests significant weaknesses in verbal memory, learning, attention, language abilities, visuospatial processes, working memory and executive function, which may result in poor psychosocial functioning and adaptation. A previous study by Colgecen et al. [16] also showed significant cognitive impairment in patients with psoriasis, measured by the Montreal Cognitive Assessment (MOCA) screening scale. Similar observations were reported in an Italian study performed in psoriasis outpatients and healthy controls [4]. The results showed significant impairment in most cognitive domains, including executive functions, verbal memory, attention and language. Psoriasis patients performed poorly on most of the neuropsychological tests, compared to the healthy subjects, and presented with more intense anxiety and depressive symptoms. In our study, about 50% of patients showed mild or moderate levels of depressive symptoms on the HADS scale. Interestingly, cognitive decline was not associated with the intensity of depressive symptoms, except psychomotor slowness. Most of the data from affective disorders show a significant correlation between cognitive decline and the severity of depression, and the persistent character of executive function impairment has also been detected in the remission period [26,27]. This may indicate an etiology of cognitive dysfunction in psoriasis that is different to that in affective disorders. Higher scores on the PASI scale assessing the severity of psoriasis were associated with a higher intensity of depressive symptoms and psychomotor slowness on the TMT A. This confirms a previous hypothesis linking cognitive decline, depression and psoriasis, likely through the chronic inflammation process [26]. 

Cognitive decline in patients with psoriasis is connected with age. Older patients show a decrease in cognitive abilities in all domains. Moreover, MRI data suggest these patients have a higher risk of neurodegenerative changes, which may result in a higher risk of cognitive disorders. In subjects with certain chronic diseases and in advanced age, a more severe atrophy of the brain can be expected, which should manifest as a reduction in its volume. Similarly, there may also be a progression of hyperintense white matter lesions in MRI FLAIR sequences, which in turn translates into a higher grade on the Fazekas scale, and also a higher WML count and WML volume in the automated segmentation results. Most of the patients with psoriasis had no or only low amounts of WML on the MRI (Fazekas 0 or 1), only a few had a moderate amount of WML (Fazekas 2) and no patients presented with extensive WML (Fazekas 3). Since small and infrequent hyperintensive changes in white matter are often incidentally found in healthy individuals, especially beginning in their fourth to fifth decade of life, “Fazekas 1” does not necessarily clearly indicate pathological processes. Admittedly, the severity of the brain changes on MRI was not very severe (Fazekas scale 0–2); however, these changes were more severe compared to healthy subjects. In addition, the MRI parameters were found to be correlated with cognitive function. It has been shown that the volume of lesions in the white matter and higher scores on the Fazekas scale (which are a subjective reflection of the extent of WML) indicate a greater severity of changes in the brain (likely mainly of a vascular nature); thus, worse results on neurological tests were in fact to be expected.

We observed a correlation between worse performance on the neuropsychological tests and a higher WML volume and periventricular WML on the Fazekas scale, as well as a higher deep WML volume, Fazekas deep WML volume and Fazekas periventricular WML scoring. In addition, better performance on verbal and executive functions was associated with higher brain volume and grey matter volume and lower scores of deep WML (i.e., less extensive changes in the white matter). This indicates that the less extensive the changes in the white matter (smaller volume of WML) and the higher the brain volume and grey matter (less atrophy), the better the expected results in cognitive tests. More robust associations were found between the neuroimaging variables and performance on the Rey AVLT. It is likely that verbal learning and memory may be recognized as sensitive parameters to the level of brain changes in patients with psoriasis. These results show the coexistence of disorders that could potentially occur in a logical sequence of events: the acceleration of brain atrophy (smaller brain volumes) and the occurrence of more extensive pathological changes in white matter (WML volume; Fazekas score) on MRI scans—likely due to vascular processes—finally results in the impairment of cognitive functions. Other studies have also suggested a relationship between the level of brain atrophy and cognitive decline in psoriasis. Kynast et al. also found relationships between white matter hyperintensities and small vessel dysfunctions and the impairment of attention, memory and social cognition [25]. These findings indicate that that brain atrophy and white matter lesions are progressive, and cognitive deterioration intensifies with the progression of brain changes. Processing speed, memory and social cognition may be recognized as neuropsychological markers of the progression of brain changes. In our study, one indicator of brain damage may have been a weakening of verbal memory and learning, especially deferred memory. 

However, not all studies are in agreement with the results obtained in this study [28]. The main reasons for this are different methodological assumptions and the diverse selection of control groups. Pezollo et al. reported no association between cognition, brain imaging markers and the risk of dementia [16]. It should be noted, however, that the control group consisted of subjects that presented other serious somatic and psychiatric diseases, such as hypertension, depression, diabetes, hypercholesterolemia and obesity, in which cognitive impairment and structural and functional brain abnormalities are well documented [29,30,31]. In addition, the authors divided the study group into patients with and without psoriasis, which indicates that the control group did not consist of healthy persons.

## 5. Limitation

This was a cross-sectional study; thus, a follow-up assessment of the same subjects with psoriasis is needed to draw reliable conclusions. To verify the relationship between MRI correlates, vascular abnormalities (especially small vessel changes in the brain) and cognitive changes, it would be worthwhile to perform an ultrasound examination of brain vessels. Another limitation of our study is the fact that it was not possible to compare neuroimaging parameters in psoriasis patients and healthy controls, due to the lack of approval from the bioethics committee to perform MRI tests in people without indications for such a procedure.

## 6. Conclusions

Our results indicate the possibility of progressive brain atrophy related to cognitive decline in psoriasis. Depression and older age can be considered as factors that exacerbate these processes in psoriasis patients. An intermediary factor may be the intensification of vascular lesions caused by changes in the white matter. 

## Figures and Tables

**Figure 1 ijerph-19-05687-f001:**
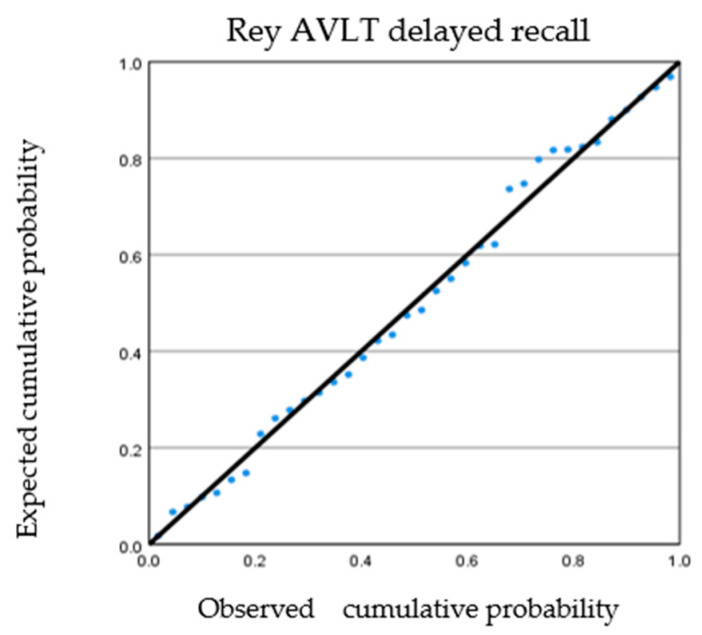
Regression plot of standardized residues for variable AVLT delayed recall.

**Table 1 ijerph-19-05687-t001:** The results of neuropsychological tests in healthy controls (group 0; *N* = 36) and patients with psoriasis (group 1; *N* = 53). The mean values, standard deviations and significance of differences between groups are also shown.

	Group	Mean ± SD	Student’s *t*-Test Significance (Two-Sided)	Median Value; IQR 25–75%	Mann-Whitney U Test Significance (Two-Sided) *p*<
**TMT A**	0	21.80 ± 6.14	0.00001	21; 17–25	0.00002
	1	30.50 ± 9.58		30; 25–38	
**TMT B**	0		No assumptions	56; 46–78	0.00521
	1			68; 54–108	
**DS forward**	0		No assumptions	6; 5–8	0.08130
	1			5; 4–7	
**DS backward**	0		No assumptions	5; 4–7	0.67653
	1			5; 4–6	
**RAVLT 1**	0	7.11 ± 1.97	0.04569	7; 6–8	0.03398
	1	6.17 ± 2.09		6; 4–7	
**RAVLT 2**	0	10.14 ± 2.13	0.00013	10; 9–12	0.00032
	1	7.93 ± 2.59		8; 6–10	
**RAVLT 3**	0	11.29 ± 2.08	0.00352	12; 10–13	0.00288
	1	9.57 ± 2.78		10; 8–11	
**RAVLT 4**	0	12.37 ± 1.90	0.00065	13; 11–14	0.00296
	1	10.36 ± 3.01		11; 8–13	
**RAVLT 5**	0	12.57 ± 1.96	0.00990	13; 11–14	0.03062
	1	11.07 ± 2.98		11; 9–14	
**RAVLT** **delayed recall**	0	11.74 ± 2.70	0.00345	12; 10–14	0.00855
	1	9.50 ± 3.79		10; 6–13	
**SCWIT A**	0		No assumptions	22; 21–25	0.08998
	1			24; 22–31	
**SCWIT B**	0		No assumptions	50; 44–65	0.00304
	1			57; 52–74	
**VF**	0	35.89 ± 9.45	0.09621	36; 31–43	0.04418
	1	31.90 ± 11.00		32; 26–39	

**Table 2 ijerph-19-05687-t002:** The results of MRI parameters in patients with psoriasis. Mean values, standard deviations and significance of differences between groups are shown.

Variable	Mean ± SD	95% Confidence Interval Min/Max		Mean Standard Error
**BV (cm^3^)**	1,000,093 ± 87,820	966,688	1,033,500	16,308
**CV (cm^3^)**	134,879 ± 12,604	130,084	139,673	2341
**GMV (cm^3)^**	698,322 ± 6518	676,823	719,820	10,495
**WML (cm^3^)**	459,360 ± 50,890	440,003	478,718	9450
**Total WML count**	80,100 ± 56,878	58,470	101,740	10,562
**Total WML volume (cm^3^)**	3644 ± 3854	2178	5111	0.716
**Deep WML count**	8210 ± 8411	5010	11,410	1562
**Deep WML volume (cm^3^)**	0.117 ± 0.169	0.052	0.181	0.031
**Fazekas deep WML**	0.450 ± 0.572	0.230	0.670	0.106
**Periventricular WML count**	0.790 ± 0.620	0.560	1030	0.115
**Periventricular WML volume (cm^3^)**	10,790 ± 5666	8640	12,950	1052
**Fazekas periventricular WML**	2231 ± 3311	0.971	3491	0.615

**Table 3 ijerph-19-05687-t003:** Pearson correlation between age, severity of psoriasis and depression using the Pearson coefficient and two-sided significance.

Age+	R	*p*=
**BV**	−0.44	0.020
**CV**	−0.51	0.005
**GMV**	−0.52	0.004
**Periventricular WML count**	0.43	0.019
**Fazekas periventricular WML**	0.43	0.020
**TMT A**	0.52	0.004
**TMT B**	0.50	0.001
**RAVLT 1**	−0.53	0.003
**RAVLT 2**	−0.57	0.001
**RAVLT 3**	−0.46	0.012
**RAVLT 4**	−0.63	0.000
**RAVLT 5**	−0.58	0.001
**RAVLT delay recall**	−0.55	0.002

**Table 4 ijerph-19-05687-t004:** Pearson correlation between MRI parameters and cognitive test performance in patients with psoriasis.

		BV (cm^3^)	CV (cm^3^)	GMV (cm^3^)	Total WML Volume (cm^3^)	Deep WML Volume (cm^3^)	Fazekas Deep WML	Periventricular WML Volume (cm^3^)	Fazekas Periventricular WML
**TMT A**	R=	−0.28	−0.19	−0.28	0.29	0.41 *	0.28	0.33	0.50 **
	*p*=	0.14	0.32	0.14	0.12	0.03	0.15	0.08	0.01
**TMT B**	R	−0.30	−0.26	−0.35	0.24	0.48 **	0.30	0.24	0.29
	*p*=	0.12	0.18	0.07	0.21	0.01	0.12	0.21	0.12
**DS forward**	R	0.07	0.10	0.16	−0.18	−0.12	−0.13	−0.15	−0.23
	*p*=	0.74	0.60	0.40	0.36	0.52	0.50	0.45	0.24
**DS backward**	R	0.35	0.34	0.39 *	−0.15	−0.07	0.02	−0.08	−0.28
	*p*=	0.07	0.08	0.05	0.44	0.74	0.93	0.68	0.14
**AVLT 1**	R	0.41 *	0.35	0.44 *	−0.37	−0.40 *	−0.34	−0.43 *	−0.541 **
	*p*=	0.03	0.07	0.02	0.05	0.03	0.08	0.02	0.00
**AVLT 2**	R	0.32	0.32	0.35	−0.45 *	−0.45 *	−0.41 *	−0.57 **	−0.49 **
	*p*=	0.12	0.09	0.07	0.02	0.01	0.03	0.00	0.01
**AVLT 3**	R	0.35	0.27	0.36	−0.38 *	−0.47 *	−0.46 *	−0.50 **	−0.49 **
	*p*=	0.06	0.16	0.05	0.04	0.01	0.01	0.01	0.01
**AVLT 4**	R	0.41 *	0.47 **	0.45 *	−036	−0.21	−0.36	−0.48 **	−0.50 **
	*p*=	0.03	0.01	0.02	0.05	0.27	0.06	0.01	0.01
**AVLT 5**	R	0.27	0.31	0.31	−0.42 *	−0.35	−0.41 *	−0.53 **	−0.54 **
	*p*=	0.15	0.10	0.11	0.03	0.06	0.03	0.00	0.00
**AVLT delay**	R	0.38 *	0.39 *	0.41 *	−0.39 *	−0.31	−0.34	−0.51 **	−0.55 **
	*p*=	0.04	0.04	0.03	0.04	0.11	0.07	0.01	0.00
**SCWIT A**	R	−0.05	−0.02	−0.16	0.28	0.60 **	0.37 *	0.22	0.43 *
	*p*=	0.79	0.92	0.42	0.15	0.00	0.05	0.25	0.02
**SCWIT B**	R	−0.28	−0.22	−0.29	−0.02	0.19	0.16	0.05	0.09
	*p*=	0.14	0.26	0.12	0.93	0.32	0.42	0.80	0.64
**VF**	R	0.48 **	0.34	0.48 **	−0.25	−0.49 **	−0.36	−0.37	−0.25

* Marked correlations are statistically significant *p* < 0.05. ** Marked correlations are statistically significant *p* < 0.01.

## Data Availability

The data presented in this study are available on request from the corresponding author (L.M.-J.). The data are not publicly available due to privacy or ethical restrictions.

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
