# Peer review of "Cognitive Functions Associated with Brain Imaging Markers in Patients with Psoriasis"

_ijerph, 2022, doi:10.3390/ijerph19095687_

Round 1

Reviewer 1 Report

This cross-sectional study  investigating the possible association between psoriasis, magnetic resonance imaging-MRI markers and cognitive impairment is very interesting and well designed.
The paper is well written in a understandable. Methods are well described. Results are appropriately discussed.

I would ask the Authors only to specify the unit of measurement of the current episode of psoriasis (months?):  "The duration of current episode of psoriasis was 7.64 ± 4.35".

Furthermore,  it should be specified:  1) if there were significant differences from the comparison of the two samples by age 2) the scores reported by the controls on the depression scale.

Author Response

Answer to the reviewers’ comments:

We would like to thank the reviewers for their valuable comments. The article has been corrected according to the suggestions of the reviewers. The corrections in the text are marked in red.

Reviewer 1

  1. I would ask the Authors only to specify the unit of measurement of the current episode of psoriasis (months?): "The duration of current episode of psoriasis was 7.64 ± 4.35".

Answer: The duration of the current episode of psoriasis was 4-9 weeks, mean 7.64 ± 4.35 weeks. This information is included on page 2.

  1. Furthermore, it should be specified:  1) if there were significant differences from the comparison of the two samples by age 2) the scores reported by the controls on the depression scale.

Answer:

1) No significant differences on age between psoriasis and the control group were found  (mean: age of patients 46.00 ± 14.255 vs. mean of controls 43.72 ± 10.737, p = 0.393, t-student test; median value and 25-75% of patients 46 (34- 58) vs. controls 44 (34-56), p=0.398 Mann-Whitney U test. This information is included on page 2.

2) The Hospital Anxiety and Depression Scale (HADS) was used only in the group of psoriasis patients, because the HADS scale is intended for the study of hospitalized subjects. One of the inclusion criteria in the control group of healthy people was the absence of symptoms of depression, anxiety and other psychiatric disorders. A correction was made in a part of the description of the studied groups on page 2.

Reviewer 2 Report

The study shows an understandable background. However, it contains several critical points e.g. unclear definition of the variables, a lack of text consistency and the obvios spelling mistakes.

Author Response

Answer to the reviewers’ comments:

We would like to thank the reviewers for their valuable comments. The article has been corrected according to the suggestions of the reviewers. The corrections in the text are marked in red.

Reviewer 2

  1. The study shows an understandable background. However, it contains several critical points e.g.

    a. unclear definition of the variables,

Answer:  The required corrections have been made to the description of neuropsychological variables in part 2.3.3. “Neuropsychological Testing” and in the tables.

b. a lack of text consistency and obvious spelling mistakes.

Answer: Text proofread was performed by a native speaker to improve the linguistic transmission of the text and eliminate language errors.

(English Editing ID MDPI: 43507).

Reviewer 3 Report

The authors compared the neuropsychological outcomes of patients with psoriasis with similar outcomes in healthy controls. Although the study shows interesting and exciting results, it has a number of serious shortcomings. Minor problems: Methods: lack of patient flowchart - The reader has no information about the screening process, how many patients dropped out and why they dropped out of the study. was the dermatologist who examined the patients blind to the neuropsychological findings?   Results: The results were also discussed in the Results section, in many cases, and this should take place in the Discussion section. Abbreviations are omitted in all table descriptions, Table 2 is described below Table 1. No such notation is used: p = 0.000, instead, for example, p <0.001 Major problem: Table 1 compares the neuropsychological outcomes of patients with psoriasis and healthy subjects. In Table 2, however, I only see the MRI results of patients with psoriasis. I don’t understand what the control group makes sense if not all of the test methods are used in the comparison. A psychological test was also performed for healthy subjects, but MRI was performed only for patients, although important conclusions could have been drawn when comparing the two groups.

Author Response

Answer to the reviewers’ comments:

We would like to thank the reviewers for their valuable comments. The article has been corrected according to the suggestions of the reviewers. The corrections in the text are marked in red.

Comments and Suggestions for Authors

The authors compared the neuropsychological outcomes of patients with psoriasis with similar outcomes in healthy controls. Although the study shows interesting and exciting results, it has a number of serious shortcomings.

1. Minor problems: Methods: lack of patient flowchart - The reader has no information about the screening process, how many patients dropped out and why they dropped out of the study.

Answer: Initially, 55 psoriasis patients were recruited to the study, and 2 of them dropped out: patient 1 because of impossibility to perform the Color–Word Interference Test (colorblindness), and patient 2 due to discharge from the hospital before the start of the examination. This information was included in part 2.1. “Study Design”

2. was the dermatologist who examined the patients blind to the neuropsychological findings?

Answer: The dermatologist who performed the initial examination of the patients was blind to the neuropsychological tests performed by the neuropsychologist and the neuroimaging data analyzed by the radiologist. This information was included in part 2.1. “Study Design”.

3. Results:

a. The results were also discussed in the Results section, in many cases, and this should take place in the Discussion section.

Answer: In accordance with the reviewer's recommendation, fragments of the text discussing the results were moved to the “Discussion".

b. Abbreviations are omitted in all table descriptions.

Answer:  In accordance with the reviewer's recommendation, all abbreviations have been corrected in the "Methods" section and in the tables.

c. Table 2 is described below Table 1.

Answer: We changed the place of the description of the results in accordance with the reviewer's recommendation.

d. No such notation is used: p = 0.000, instead, for example, p <0.001

Answer: In the tables, the p values were changed to three decimal places.

e. Major problem: Table 1 compares the neuropsychological outcomes of patients with psoriasis and healthy subjects. In Table 2, however, I only see the MRI results of patients with psoriasis. I don’t understand what the control group makes sense if not all of the test methods are used in the comparison. A psychological test was also performed for healthy subjects, but MRI was performed only for patients, although important conclusions could have been drawn when comparing the two groups.

Answer: The control group was a reference for the neuropsychological tests.  Unfortunately, the bioethics committee did not agree to perform an MRI examination in healthy people without medical indications for such a medical diagnostic procedure. This is described in the project limitations.

Text proofread was performed by a native speaker to improve the linguistic transmission of the text and eliminate language errors.  (English Editing ID MDPI: 43507).

Round 2

Reviewer 2 Report

Accept in present form

Reviewer 3 Report

accepted